# Optimizing the Composition of the Substrate Enhances the Performance of Peroxidase-like Nanozymes in Colorimetric Assays: A Case Study of Prussian Blue and 3,3′-Diaminobenzidine

**DOI:** 10.3390/molecules28227622

**Published:** 2023-11-16

**Authors:** Pavel Khramtsov, Artem Minin, Zarina Galaeva, Elena Mukhlynina, Maria Kropaneva, Mikhail Rayev

**Affiliations:** 1Institute of Ecology and Genetics of Microorganisms, Urals Branch of RAS, 614081 Perm, Russia; kropanevamasha@gmail.com (M.K.); mraev@iegm.ru (M.R.); 2Biology Faculty, Perm State University, 614990 Perm, Russia; galaevazarina@gmail.com; 3M.N. Mikheev Institute of Metal Physics Urals Branch of RAS, 620108 Ekaterinburg, Russia; calamatica@gmail.com; 4Faculty of Biology and Fundamental Medicine, Ural Federal University, 620002 Ekaterinburg, Russia; 5Institute of Immunology and Physiology, Urals Branch of RAS, 620049 Ekaterinburg, Russia; elena.mukhlynina@yandex.ru

**Keywords:** Prussian Blue, immunoassay, peroxidase, immunohistochemistry, Western blotting, dot blot

## Abstract

One of the emerging trends in modern analytical and bioanalytical chemistry involves the substitution of enzyme labels (such as horseradish peroxidase) with nanozymes (nanoparticles possessing enzyme-like catalytic activity). Since enzymes and nanozymes typically operate through different catalytic mechanisms, it is expected that optimal reaction conditions will also differ. The optimization of substrates for nanozymes usually focuses on determining the ideal pH and temperature. However, in some cases, even this step is overlooked, and commercial substrate formulations designed for enzymes are utilized. This paper demonstrates that not only the pH but also the composition of the substrate buffer, including the buffer species and additives, significantly impact the analytical signal generated by nanozymes. The presence of enhancers such as imidazole in commercial substrates diminishes the catalytic activity of nanozymes, which is demonstrated herein through the use of 3,3′-diaminobenzidine (DAB) and Prussian Blue as a model chromogenic substrate and nanozyme. Conversely, a simple modification to the substrate buffer greatly enhances the performance of nanozymes. Specifically, in this paper, it is demonstrated that buffers such as citrate, MES, HEPES, and TRIS, containing 1.5–2 M NaCl or NH_4_Cl, substantially increase DAB oxidation by Prussian Blue and yield a higher signal compared to commercial DAB formulations. The central message of this paper is that the optimization of substrate composition should be an integral step in the development of nanozyme-based assays. Herein, a step-by-step optimization of the DAB substrate composition for Prussian Blue nanozymes is presented. The optimized substrate outperforms commercial formulations in terms of efficiency. The effectiveness of the optimized DAB substrate is affirmed through its application in several commonly used immunostaining techniques, including tissue staining, Western blotting assays of immunoglobulins, and dot blot assays of antibodies against SARS-CoV-2.

## 1. Introduction

Enzymes play a crucial role in amplifying signals in modern diagnostic techniques such as the enzyme-linked immunosorbent assay (ELISA), lateral flow assays, Western blotting, immunoblotting, immunohistochemistry, and various biosensor-based techniques [1]. In colorimetric assays, enzymes catalyze the conversion of substrates into brightly colored products that can be easily detected either visually or with the aid of spectrophotometers and scanners. Horseradish peroxidase (HRP) is one of the most popular enzyme labels in commercial colorimetric assays [2]. Currently, there are attempts to replace HRP with nanoparticles that mimic its catalytic activity [3,4]. These nanoparticles, known as nanozymes, typically consist of transition metal compounds, noble metals, or carbon allotropes. The mechanism of action of these nanomaterials can be completely different from that of HRP and often involves the generation of oxygen radicals [5,6]. However, the results of their application are ultimately the same: the peroxide-dependent oxidation of colorless substrates into colored products.

Some analytical applications use special chromogenic substrates, known as precipitating substrates, that generate colored insoluble products which precipitate at the site of the enzymatic reaction [7]. These substrates allow for the determination of not only the quantity but also the location of an analyte on a membrane or histological section. This is crucial for techniques such as ELISPOT [8], tissue staining [9], blotting techniques [10], and paper-based assays [11]. Several precipitating substrates are used in colorimetric assays, including 3,3′-diaminobenzidine (DAB), 3-Amino-9-ethylcarbazole (AEC), and 4-Chloro-1-naphthol (4-CN). Since this article focuses on DAB, it is important to note that in the presence of peroxide, HRP converts this substrate into a polymerized brown or reddish insoluble product. Immunostaining methods using HRP and DAB have been known for many years, resulting in the availability of numerous commercially optimized substrate formulations with well-optimized pH values, chemical compositions [12,13], and the presence of enhancers such as imidazole [12,13,14,15] or bi- and trivalent metal cations [16].

In articles that describe colorimetric assays based on the nanozyme–DAB detection system, the compositions of the DAB substrate solution are rarely reported [17,18,19,20]. Usually, authors use substrates from commercial kits specifically designed for HRP. However, there is a growing body of evidence indicating that not only the pH but also the composition of substrate solutions can affect the oxidoreductase-like activity of nanozymes [21,22,23,24]. Moreover, the effect of the buffer varies among different chromogenic substrates [25]. Therefore, optimizing the composition of the substrate can be an effective and cost-efficient approach to enhancing the detection limits of an assay, but this option is mostly overlooked in modern research. Our literature search yielded a limited number of papers that focused on optimizing the substrate buffer in colorimetric nanozyme-based assays. Some of these reports are summarized below. Hormozi-Jangi et al. [21] demonstrated the significantly higher efficiency of a DAB substrate prepared using an acetate buffer compared to citrate, TRIS, and phosphate buffers in a MnO_2_-based assay of triacetone triperoxide. In contrast, another study [26] indicated that the effect of the acetate buffer was nearly identical to that of the phosphate and citrate buffers in the V_2_O_5_-based colorimetric assay. The influence of the buffer was also observed in laccase-mimicking copper-based nanozymes utilized for phenol identification [27].

In this study, we highlight the significance of the DAB substrate buffer composition, in addition to its pH, when utilizing nanozymes as labels in colorimetric assays. For this study, as an alternative to horseradish peroxidase (HRP), we used Prussian Blue nanozymes synthesized following the method outlined by Komkova et al. [28]. These nanozymes exhibit excellent peroxidase-like activity and can be a promising substitute for HRP. Interestingly, the addition of salts such as NaCl or NH_4_Cl significantly enhances the intensity of the staining by Prussian Blue. Conversely, the enhancers commonly found in commercial DAB substrate solutions, such as metal cations or imidazole, do not exhibit the same level of efficacy and may even diminish the color signal intensity produced by nanozymes. The optimized DAB substrate was applied in Western blotting assays for mouse and human IgG, as well as in dot blot assays for human antibodies against the spike protein of SARS-CoV-2. It is worth noting that this study represents the first demonstration of Prussian Blue nanozymes being employed as peroxidase-like labels in immunohistochemistry and Western blotting.

## 2. Results and Discussion

### 2.1. Characterization of Nanoparticles

Prussian Blue nanoparticles were synthesized by reducing FeCl_3_/K_3_[Fe(CN)_6_] with H_2_O_2_ in the presence of citric acid [29]. Then, the nanoparticles were coated with a gelatin A layer, to which streptavidin (PB@Gel/Str) or BSA (PB@Gel/BSA, control nanoparticles) was covalently attached. The nanoparticle dispersions exhibited a blue color and displayed the characteristic absorbance spectrum of Prussian Blue (Figure 1a).

The average hydrodynamic diameter of PB@Gel/Str was 230 nm (Figure 1b). TEM analysis revealed that the nanoparticles had an irregular shape and consisted of tightly aggregated smaller nanoparticles (Figure 1i,j). Despite this, the polydispersity index was 0.107, as determined using DLS for the PB@Gel/Str sample, indicating a relatively narrow size distribution. This is likely due to the well-known bias of dynamic light scattering (DLS) towards detecting larger particles. Smaller particles scatter light less strongly, making them more challenging to detect and not significantly impacting the polydispersity index value. Additionally, the DLS method assumes that the measured objects are spherical in shape. For non-spherical objects with specific diffusion coefficients, the method provides the size of an equivalent sphere. As a result, the obtained hydrodynamic diameters are larger than the actual size of the nanoparticles but still offer a relatively accurate estimate of their size.

The zeta potential of PB@Gel/Str above pH 5 was negative (Figure 1c). The zeta potential of the nanoparticles is influenced by the isoelectric point of gelatin A, which serves as the coating. The isoelectric point of gelatin A typically falls within the pH range of 7 to 9. However, treatment with glutaraldehyde removes lysine amino groups from the gelatin layer, shifting the isoelectric point of the aldehyde-treated gelatin A to a pH range of 4 to 5. Given that the pKa of DAB is 5.5 (predicted by MolGpKa [30]), the electrostatic adsorption of DAB is more likely to occur at pH 5–6, where the nanoparticles and DAB have opposite charges. The electrostatic attraction of substrate molecules in some cases facilitates higher catalytic activity at certain pH [31]. However, distinct relationships were not observed.

The PB@Gel/Str and PB@Gel/BSA conjugates were stored as aqueous dispersions, and their size and polydispersity remained unchanged for over 20 weeks (Figure 1g). Prussian Blue is sensitive to hydrolysis under neutral conditions [32], and gelatin-coated Prussian Blue is unstable even when stored in acidic buffers [29]. Short-term stability in physiological and weakly acidic conditions is highly desirable for immunostaining applications. No aggregation was detected at pH 5–7, whereas at pH 4, the diameter of the nanoparticles increased dramatically, indicating aggregation. An acidic buffer is typically added at its final stage as a component of the substrate solution during a colorimetric assay. At this point, the nanoparticles are already attached to the solid phase (surface of paper, polystyrene well, or tissue section) and cannot aggregate. Therefore, PB@Gel/Str can be stored for several months in deionized water as concentrated suspensions, while their stability in buffers is sufficient for practical applications. The different stability of PB@Gel/Str at pH 4 and pH 5–7 could have potentially affected the results of our buffer optimization experiments, which were performed in solution. Therefore, additional tests of the best substrates in solid-phase assays to mitigate any potential aggregation effects were conducted.

Both PB@Gel/Str and PB@Gel/BSA nanoparticles catalyzed the oxidation of TMB and DAB in the presence of H_2_O_2_, whereas no color development was observed in samples without H_2_O_2_ (Figure 1d,e, Appendix A and Appendix A). These results confirm that Prussian Blue has peroxidase-like activity but lacks oxidase-like activity, which is consistent with previous reports [28,33].

The specific binding ability of PB@Gel/Str to biotinylated targets was confirmed by the direct detection of Bi-BSA adsorbed onto polystyrene plates and nitrocellulose strips. Upon the addition of HRP-Str or PB@Gel/Str, color development was observed in wells containing adsorbed Bi-BSA, with the intensity of the color being proportional to the concentration of Bi-BSA (Figure 1f and Appendix A). The control nanoparticles, PB@Gel/BSA, did not produce any signal. The same results were observed on nitrocellulose strips (Figure 1h). The binding capacity of biotin was quantitatively measured using 6-Carboxyfluorescein-labeled biotinylated oligoDNA. At a ratio of 625:1 (pmol:mg) of oligoDNA to nanoparticles, PB@Gel/Str exhibited a binding capacity of 241 pmol/mg, whereas PB@Gel/BSA had a binding capacity of only 10 pmol/mg. The binding of non-biotinylated oligoDNA was 8-fold lower (Appendix A).

### 2.2. DAB Substrate Optimization

Staining using DAB has been widely used in immunohistochemistry and other immunological techniques for nearly fifty years. Over this time, the composition of the DAB substrate solution has been carefully optimized. Commercial peroxidase substrate preparations contain enhancers such as imidazole and stabilizers such as polymers and reducing agents to prevent the spontaneous oxidation of DAB during storage [12,13,14]. However, these optimizations have been specifically designed for peroxidase, which has different catalytic properties and mechanisms compared to nanozymes. Prussian Blue nanozymes, for example, do not produce hydroxyl radicals at acidic pH but instead scavenge them [34,35]. According to one study [34], the peroxidase-like activity of Prussian Blue involves the oxidation of Prussian Blue by H_2_O_2_ to Berlin Green or Prussian Brown, which then oxidizes TMB. However, an alternative catalytic mechanism of Prussian Blue was demonstrated in another paper [36]: firstly, Prussian Blue nanozymes are reduced to Prussian White by TMB, and then they are oxidized by H_2_O_2_. These different modes of action from HRP [2] suggest that the optimal reaction conditions for HRP and Prussian Blue may also differ. Additionally, nanozymes require higher concentrations of H_2_O_2_ compared to natural peroxidases. With these considerations in mind, we aimed to optimize the composition of the DAB substrate to achieve high color intensity and low background staining in immunosorbent assays.

The catalytic activity of Prussian Blue and HRP at various pH levels in different buffers was compared. The oxidation degree was measured by monitoring the absorbance at 470 nm, while absorbance at 800 nm (turbidity) indicated the formation of DAB polymer species. In general, the effect of buffer composition on DAB oxidation was similar for both catalysts (Figure 2). However, when compared to Prussian Blue, peroxidase produced higher absorbance values at pH 4 and 5 but not at pH 6 and 7. This suggests that the difference in the color intensity of the reaction product is influenced by the buffer composition rather than the concentration of the catalysts.

The most significant finding is that not only the pH but also the type of buffer has an essential impact on the DAB oxidation for both Prussian Blue and HRP.

In the process of optimizing the substrate, we manipulated various parameters, including buffer type, pH, and molarity, as well as the concentrations of DAB, H_2_O_2_, and additives such as salts, transition metal cations, and imidazole. DAB is a precipitating substrate that forms a dark brown insoluble sediment upon oxidation. Therefore, two parameters were evaluated: the rate of DAB oxidation (A_470_) and polymerization (A_800_). The optimization process was conducted in 96-well plates and involved several steps. Initially, we identified conditions that led to an increase in both A_470_ and A_800_, without inducing the spontaneous oxidation of DAB in the absence of nanozymes. The complete list of tested variables can be found in Appendix A. Subsequently, we retested these variables individually and in combinations (e.g., the addition of NaCl with increased DAB concentration). Finally, the best combinations were selected and evaluated in a colorimetric dot blot assay of Bi-BSA using white polystyrene plates. Three of the most promising substrates were chosen for comparison with commercial ones.

The addition of NaCl or NH_4_Cl up to 1.5 M resulted in enhanced A_470_ and A_800_ for most of the tested buffers, while KCl had no effect (Figure 3 and Appendix A). Substrates containing Na_2_SO_4_ formed a white sediment prior to the addition of nanozyme. Increasing the buffer molarity from 33 mM to 100 mM either had no influence or had negative effects on most of the buffers (Appendix A). Higher concentrations of DAB generally led to higher absorbances but also increased the spontaneous oxidation of DAB. However, for the MES and Na-citrate buffers at pH 6 and 7, an increase in DAB concentration to 2 or 3 mg/mL resulted in higher absorbance without a significant effect on the spontaneous oxidation of DAB (Appendix A). Concentrations of H_2_O_2_ above 0.1% provided negligible increases in color intensity (Appendix A). Additionally, TRIS buffers with pH levels up to 8.0 were tested (Appendix A). At a pH of 7.4, turbidity reached its maximum level, while A_470_ only slightly declined. Notably, Hormozi Jangi et al. recently reported the strong enhancement effect of an acetate buffer [21], but in our experimental setup, the acetate buffer yielded only moderate color intensity (Appendix A).

Transition metal cations are widely known as enhancers for DAB staining in HRP-based techniques, but we found them to be ineffective. Based on the literature, we selected several cations (Fe^3+^, Mn^2+^, Co^2+^, Cu^2+^, Ni^2+^) [16]. Cupric and ferric cations increased absorbance at 470 and 800 nm, as shown in Figure 4a,b. However, this effect can be explained by their inherent peroxidase-like activity, because the formation of a brown product was observed even before the addition of nanozymes (spontaneous DAB oxidation), as depicted in Figure 4c,d. Another enhancer for DAB, imidazole, is commonly found in commercial substrate formulations (according to patent search). We discovered that adding imidazole at 10 mM, which is considered optimal for HRP [14], actually decreases both A_470_ and A_800_ with the use of PB@Gel/Str (Figure 5).

Overall, the addition of salts proved to be the most effective method for enhancing the signal. The enhanced effect of the high salt concentration could be attributed to a salting-out effect or a decrease in the buffer pH [37]. The effect of salt addition on various pH levels of the same buffer was explored. This research helped us to find that the turbidity increasing was not a result of the pH changing. Therefore, we believe that the salting-out effect, rather than a change in pH caused by high salt concentration, is responsible for the stronger signal. Salts can also affect the substrate’s affinity towards the catalyst or accelerate DAB polymerization/aggregation, leading to higher turbidity values (A_800_).

Based on the aforementioned experiments, for a comparative study, 30 different substrate compositions (Appendix A) were formulated. Solutions with varying concentrations of Bi-BSA were dotted onto wells of white opaque polystyrene plates, and the adsorbed Bi-BSA was detected using PB@Gel/Str and DAB substrates (Appendix A). To measure the signal intensity, images of the wells after the assay were captured and analyzed using ImageJ software (https://imagej.net/ij/index.html, accessed on 22 June 2023). However, due to the non-homogeneous coloring of the spots, the results of the image analysis were inaccurate. Therefore, the best substrates were selected based on both image analysis and a visual assessment of the wells (Appendix A). Consistent with the intensity of the colored spots in the immunoassay, we also evaluated the spontaneous oxidation of the DAB substrates in 96-well plates by measuring A_470_ and A_800_ ten minutes after their preparation (Figure 6d).

Confirming the previous findings, the addition of a high concentration of salt resulted in higher color intensity. Unfortunately, increasing the concentration of DAB to 2 or 3 mg/mL led to high background and the formation of a white sediment. Three substrates were chosen as the best options:A 33 mM Na-citrate with 1.5 M NaCl, pH 6, 1 mg/mL DAB; 0.1% H_2_O_2_.A 33 mM Na-citrate with 1.5 M NH_4_Cl, pH 7, 1 mg/mL DAB; 0.1% H_2_O_2_;A 30 mM HEPES–HCl, 3 mM Na-citrate, with 1.5 M NaCl, pH 7, 1 mg/mL DAB; 0.1% H_2_O_2_.

These substrates were compared to an HRP-optimized substrate patented by Life Corp. (formerly Invitrogen), which, according to the patent, outperforms several commercial buffers from various suppliers [15]. The HRP-optimized substrate consisted of a 200 mM Na-acetate buffer with 0.54 mg/mL DAB, 1 mM DTPA, 50 mM imidazole, and 0.03% H_2_O_2_ at pH 5. In addition to the control buffer, we also used a control buffer with an increased H_2_O_2_ concentration of 0.1%, given the suboptimal concentration of H_2_O_2_ in the control buffer for nanozymes.

In the dot blot assay of Bi-BSA, the optimized substrates showed significantly enhanced color intensity compared to the control substrates (Figure 6a,b). Moreover, there was no non-specific staining of the polystyrene surface, despite a slight degree of spontaneous DAB oxidation (Figure 6d). A comparison of the substrates in 96-well plates confirmed the results of the dot blot assay. The optimized substrates generated 2–5-fold higher absorbance values at both 470 and 800 nm (Figure 6c).

After developing DAB substrate compositions optimized for Prussian Blue nanozymes, our aim was to demonstrate their effectiveness using practical examples.

### 2.3. The Effect of Substrate Buffer in Nanozyme-Based Immunostaining

We explored the potential use of Prussian Blue nanozymes as a tag for immunohistochemical tissue staining, which, to our knowledge, has not been previously reported. We chose pancreatic islets as a model because they are morphologically distinct structures within the pancreas, which has scientific and clinical importance. Tissue sections were sequentially treated with antibodies against insulin found in the islets, biotin-labeled secondary antibodies, and either streptavidin-linked HRP or PB@Gel/Str.

As mentioned earlier, the composition of the DAB solution has a noticeable impact on the nanozyme’s activity. In the first experiment (Figure 7a,b), commercial DAB was used with both HRP and the nanozyme. It provided intense staining for HRP (Figure 7a), but no signal was detected in the case of the nanozyme (Figure 7b). Possible reasons for this could be the presence of quenching additives or a suboptimal hydrogen peroxide concentration. In the second experiment (Figure 7d–g), sections treated with PB@Gel/Str were incubated in a custom DAB solution based on a Na-citrate buffer without imidazole. This resulted in reliable staining of the islets. They were clearly distinguishable from the surrounding tissue when subjected to microscopic observation (Figure 7e–g) and through the use of image analysis software (Figure 7c).

Western blot and dot blot assays are laboratory techniques that are widely used in medicine and biology as confirmatory tests in clinical diagnostics, as well as for protein identification, the characterization of antibodies’ binding abilities, and the study of protein–target interactions. The principle of these methods is similar: protein–protein complexes, often antibody–antigen complexes, are formed on the surface of a solid support and detected through an enzyme-driven color reaction. Proteins are transferred from a gel to a porous support (in the case of Western blot) or manually spotted on a porous or nonporous support (in the case of dot blot). HRP is typically the label of choice in these applications. For the comparison of the Prussian Blue conjugates and the substrate, a simplified version of the dot blot assay was performed as described in previous sections. In this section, we present assay schemes that are closer to real laboratory applications. Additionally, to the best of our knowledge, the application of Prussian Blue nanozymes as labels in Western blotting has not been reported. For both analyses, one of three optimized DAB substrates was used: 33 mM Na-citrate with 1.5 M NaCl, pH 6; 1 mg/mL DAB; 0.1% H_2_O_2_.

We performed gel electrophoresis of mouse IgG, human IgG, BSA, and Bi-BSA in an 8% polyacrylamide gel. The proteins were then transferred from the gel to nitrocellulose membranes and treated sequentially with biotinylated Staphylococcal protein A (Bi-PtA), PB@Gel/Str, and the optimized DAB substrate. An identical gel was stained with the protein dye Coomassie Brilliant Blue. Protein A binds to both human and mouse antibodies [38], and the biotin tag facilitates its interaction with PB@Gel/Str. Bi-BSA and BSA were used as positive and negative controls, respectively. The results of the Western blotting demonstrate the successful detection of both antibodies and Bi-BSA, while no colored spot was observed in the BSA lane (Figure 8d,e).

The aim of the following experiment was to detect antibodies against the spike protein of SARS-CoV-2 in human blood serum using a dot blot assay. We analyzed pooled serum samples collected from individuals in 2014 (seronegative) and 2022 (seropositive) whose serological status regarding the presence of antibodies to the spike protein had been confirmed via the ELISA. The spike protein was dotted onto nitrocellulose test strips. Mouse IgG and Bi-BSA were used as positive controls, while BSA served as the negative control. The test strips were then incubated in serum samples (diluted at 1:100, 1:1000, and 1:10,000), Bi-PtA, PB@Gel/Str, and the optimized DAB substrate (Figure 8a). In the positive sample, brown spots were observed in the areas of the membrane where the spike protein was adsorbed, as shown in Figure 8b. In contrast, no spots were observed on the test strip incubated in the negative sample, as depicted in Figure 8c. Quantitative analysis revealed an almost three-fold increase in spot coloration in samples treated with the optimized substrate compared to the commercial substrate used in Section 3.2. This finding confirms the efficacy of the optimization procedure. Therefore, the combination of the Prussian Blue label and the optimized DAB substrate shows promise as an alternative to HRP in various immunostaining techniques.

## 3. Materials and Methods

### 3.1. Materials

Streptavidin was purchased from ProspecBio (Rehovot, Israel). Potassium hexacyanoferrate (III), casein, gelatin A 180 bloom, Proclin-950, and hydrogen peroxide were purchased from Sigma-Aldrich (Saint Louis, MO, USA). Iron (III) chloride hexahydrate, Tween-20, glutaraldehyde, ammonium chloride, citric acid, TRIS, HEPES, MES, glycine, sodium phosphate, sodium bicarbonate, 3,3′,5,5′-tetramethylbenzidine dihydrochloride (TMB), hydrogen peroxide, and glycerol were purchased from ITW (Glenview, IL, USA). Diaminobenzidine and dialysis tubing (cellulose membrane; 10,000 MWCO) were purchased from Thermo Scientific (Waltham, MA, USA). Potassium hydroxide, sodium hydroxide, sulfuric acid, and hydrochloric acid were purchased from Reakhim (Moscow, Russia). Bovine serum albumin was purchased from Biosera (Cholet, France). Oligonucleotides 5′-Biotin- GGGGCACGTTTATCCGTCCCTCCTAGTGGCGTGCCCC-FAM-3′ (Bi-D17.4-FAM) and 5′-NH_2_- GGGGCACGTTTATCCGTCCCTCCTAGTGGCGTGCCCC-FAM-3′ (NH_2_-D17.4-FAM) were obtained from Syntol (Moscow, Russia). Mouse monoclonal IgG2a (against human prostate specific antigen) and the recombinant spike protein of SARS-CoV-2 were obtained from HyTest (Turku, Finland). Additionally, 96-well polystyrene plates (high binding) were purchased from SPL Life Sciences (Pocheon-si, Republic of Korea). Nitrocellulose membrane (0.45 μm pore diameter) was purchased from Bio-Rad (Hercules, CA, USA). Horseradish peroxidase conjugated with streptavidin (HRP-Str) was purchased from Imtek (Moscow, Russia). Commercial TMB substrate buffer was obtained from an anti-Pertussis IgG ELISA kit (Euroimmun, Lübeck, Germany). The biotinylation of BSA was performed as described in [39].

The blood serum samples used were taken from our laboratory collection. Serum samples were obtained in 2014 (pre-COVID-19 era) and 2022 (post-COVID-19 era) in the framework of other projects dedicated to the study of post-vaccination immunity. Sera samples were stored at −20 °C. Antibodies against the spike protein of SARS-CoV-2 in these samples were detected using the ELISA (Vektor-Best, Novosibirsk, Russia). Positive (>4000 BAU/mL) and negative sera (no antibodies detected) were pooled and used in dot blot assays.

Instrumentation. A Multiskan Sky UV-Vis Reader was purchased from Thermo Scientific (USA). A ZetaSizer NanoZS particle analyzer was purchased from Malvern (Great Malvern, UK). A VCX-130 ultrasonic processor was purchased from Sonics & Materials (Newtown, CT, USA). The Trans-Blot^®^ SD Semi-Dry Transfer Cell was purchased from Bio-Rad (USA).

### 3.2. Synthesis of Prussian Blue Nanoparticles

Citric acid was added to a mixture of 3.125 mM FeCl_3_ and 3.125 mM K_3_[Fe(CN)_6_] to a final concentration of 2 mM. Then, H_2_O_2_ was added to a final concentration of 22 mM. The final volume of the reaction mixture was 250 mL. Synthesis took place for 60 min at +30 °C. The nanoparticles were washed three times with water via centrifugation at 16,000× *g*. Before the addition of water, nanoparticle sediments were redispersed via vortexing. After the final centrifugation, 25× *g* mL of H_2_O was added. The suspension was homogenized via an ultrasound (6 mm probe, 60% amplification, 60 min) on ice and then centrifuged at 100× *g* for 15 min to remove remaining aggregates. Supernatants were stored at +4 °C.

### 3.3. Conjugation of Prussian Blue Nanoparticles with Streptavidin

Prussian Blue nanoparticles were added to the gelatin A solution to a final concentration of 3.8 mg/mL (~10 mL). The mass ratio of nanoparticles to gelatin was 1:8. The dispersion was vortexed and sonicated (probe diameter—3 mm; amplification—60%; duration—10 s) and then kept at +37 °C for 1 h on a rotary mixer (10 rpm). Gelatin-coated Prussian Blue (PB@Gel) was dropwise added to 10 mL of 25% glutaraldehyde (pH 7, adjusted with 1 M NaOH) and then kept at +37 °C for 0.5 h. After cross-linking and after following the conjugation steps, absorbance was measured at 700 nm to assess the concentration of Prussian Blue. Glutaraldehyde-treated PB@Gel was triple-washed with water via centrifugation at 20,000× *g*. After each wash, nanoparticle pellets were redispersed via sonication (probe diameter—3 mm; amplification—60%; duration—10 s). After the final wash, 8 mL nanoparticle dispersion was prepared in a 10 mM phosphate buffer, pH 7, and then divided into 2 portions via the addition of streptavidin or BSA (100 µg per 1 mg of nanoparticles). The mixtures were vortexed and incubated on a rotary mixer (10 rpm) overnight at +4 °C. Remaining aldehyde groups were inactivated via the addition of 0.1 M glycine for 2 h at +37 °C. Nanoparticles were washed as described above and sonicated (probe diameter—3 mm; amplification—60%, duration—30 s). The conjugates were stored at +4 °C. Conjugates of PB@Gel with streptavidin and BSA were labeled as PB@Gel/Str and PB@Gel/BSA, respectively.

### 3.4. Characterization of Nanoparticles

Hydrodynamic diameter and polydispersity. Nanoparticles were diluted to 25 μg/mL in 750 μL of 0.1 M Na-citrate buffers, pH 4–7, 0.1 M Na-phosphate buffer, pH 7, or deionized water. Solvents were filtered through a 0.22 μm syringe filter prior to the addition of nanoparticles. Measurements were performed in auto mode at room temperature in 2 mL plastic cuvettes. A general purpose model was used to fit the data. Three technical replicates were performed for each sample. Intensity-weighted size distribution and the mean values and standard deviation of the z-average diameter and polydispersity index are reported.

Zeta potential. Nanoparticles were diluted to 25 μg/mL in 700 μL of 0.01 M Na-citrate buffers, pH 4–7. Measurements were performed in auto mode at room temperature in 4 mL plastic cuvettes using the Dip Cell electrode. Three technical replicates were performed for each sample. The mean values and standard deviation of the zeta potential are reported.

Size and shape. The nanoparticles were investigated via transmission electron microscopy using a CM30 SuperTWIN microscope (Philips, Amsterdam, the Netherlands). For this purpose, an aqueous suspension of nanoparticles was ultrasonicated and added to a copper mesh coated with a layer of formvar and carbon before being dried and examined via the use of a microscope.

UV-Vis absorbance spectrum. Nanoparticles were diluted to 25 μg/mL in water. Absorbance spectra were measured in a quartz cuvette with a path length of 10 mm.

Catalytic activity. Nanoparticles were added to 2 μg/mL in various substrate solutions. The DAB substrate solution was 0.1 M Na-citrate buffer, pH 5, with 1 mg/mL of DAB and 0.3% of H_2_O_2_. The TMB substrate solution was prepared by mixing 9 mL of 0.1 ammonium citrate buffer, pH 4, 1 mL of TMB (1 mg/mL) in DMSO and 100 μL of 30% H_2_O_2_. Substrate solutions were preheated at +37 °C. Measurements of absorbance at 562 nm (TMB) and 470 nm (DAB) were performed at the same temperature. Controls without H_2_O_2_ or nanoparticles were also measured.

Interaction with biotinylated targets. Polystyrene plates were used. Wells of 96-well plates were filled with 100 μL Bi-BSA (0–100 ng/mL) diluted in 0.1 M Na-carbonate buffer, pH 9.5. The plates were incubated for +4 °C overnight. The wells were washed three times with 350 μL of PBS-Tw (PBS + 0.1% of Tween-20). In each well, 200 μL of the blocking buffer (PBS-Tw + 1% BSA + 1% casein) was added. The plates were blocked for 1 h and washed. One hundred microliters of PB@Gel/Str (25 μg/mL), PB@Gel/BSA (25 μg/mL), or HRP-Str (1:500) diluted in 10 mM sodium phosphate buffer, pH 7 + 0.1% Tween-20 + 1% casein + 1% BSA were then added for 1 h. The wells were washed and filled with 100 μL of substrate solutions, namely commercial TMB solution (for HRP conjugate) or mixture of 9 mL of 0.1 M ammonium citrate buffer, pH 4, 1 mL of TMB (10 mg/mL) in DMSO, and 100 μL of 30% H_2_O_2_ (for Prussian Blue conjugates). After 30 min, 100 μL of 2 M H_2_SO_4_ was added to stop the reaction. Absorbance was measured at 450 nm using a plate reader. All incubation steps (except for coating) were performed in a thermal shaker at +37 °C at 300 RPM.

Nitrocellulose strips. Two-fold dilutions of Bi-BSA in PBS (2 μL; from 100 μg/mL to 1.56 μg/mL) were spotted onto nitrocellulose membrane strips (5 × 80 mm). After that, the strips were dried for 30 min at +37 °C and washed with 5 mL of PBS-Tw three times for 5 min; non-specific binding sites were blocked with 5 mL of PBS-Tw + 1% casein + 1% BSA solution for 60 min and washed. Next, the strips were incubated with 3.5 mL of 0.025 mg/mL PB@Gel/Str or PB@Gel/BSA in 10 mM sodium phosphate buffer, pH 7 + 0.1% Tween-20 + 1% casein + 1% BSA and incubated for 60 min and washed again with 10 mM sodium phosphate buffer, pH 7. Then, DAB solution (1 mg/mL in PBS, pH 7.2 + 0.02% H_2_O_2_) was added and incubated for 5 min. After washing, the test strips were dried and scanned on a Canoscan LiDE 600f office scanner using Canoscan Toolbox 5.0 software.

Binding of biotinylated oligoDNA. OligoDNA (Bi–D17.4–FAM and NH_2_–D17.4–FAM) was incubated for 5 min at +90 °C. After that, it was cooled to room temperature and centrifuged at 3000× *g* for 1 min. OligoDNA were diluted to 200/100/50 nM in 10 mM PBS 7.0 pH + 1% BSA + 1% Casein + 0.1% Tween-20. Then, the solution was mixed with an equal amount of 0.08 mg/mL PB@Gel/Str or PB@Gel/BSA and incubated at +37 °C for 60 min. Nanoparticles and bound oligoDNA were sedimented by centrifugation at 20,000× *g* for 15 min. The fluorescence (485/512 nm) of supernatants (100 μL) was measured in a black 96-well plate. Solutions of oligoDNA with known concentrations were used as calibrators.

### 3.5. DAB Substrate Optimization

The first part of the optimization experiments was performed in 96-well plates. Various substrate solutions in terms of pH, ionic strength, buffer molarity and composition, and H_2_O_2_ and DAB concentration were prepared; then, nanozymes were added (Table 1). Plates were incubated at +37 °C on a shaker (300 RPM). Absorbance was measured at 470 nm (color intensity of oxidized DAB) and 800 nm (turbidity) before and after the addition of the nanozymes. Various additives (e.g., salt or imidazole solutions) were added instead of water in the corresponding experiments. The starting concentrations of H_2_O_2_ and DAB were adjusted in the corresponding experiments (e.g., we added 30 μL of DAB with concentrations from 2.5 to 40 mg/mL). In most of the experiments, the molarity of the buffers was 0.1 M, giving a final molarity of 33 mM (except for the McIlvaine buffer, which was simply three times diluted). Buffers with 3 times higher molarity (0.3 M) were used to achieve a final molarity of 0.1 M when necessary. After the addition of the nanozymes, the solution in the wells was quickly mixed with the tips of a pipette.

### 3.6. Direct Assay for Bi-BSA Detection with PB@GelA/Str and PB@GelA/BSA

For this, 96-well ELISA plates were used. One hundred microliters of different concentrations of Bi-BSA (from 100 ng/mL to 1.56 ng/mL) in a 0.2 M carbonate buffer (pH 9.6) were added into the wells of 96-well polystyrene plates. The plates were kept at +4 °C overnight. The plates were washed three times with 350 of PBS with 0.1% Tween-20, pH 7.4 (PBS-Tw) using a microplate washer, and then 200 μL of a blocking buffer (PBS-Tw + 1% casein + 1% BSA, pH 7.4) was added. After 60 min of blocking, the plates were washed three times. Next, 0.025 mg/mL PB@GelA/Str or PB@GelA/BSA in 10 mM sodium phosphate buffer with 0.1% Tween-20, 1% casein, and 1% BSA, pH 7 (100 μL per well), were added; then, the plates were incubated for 60 min and washed three times with 10 mM sodium phosphate buffer, pH 7 + 0.1% Tween-20. After washing, 100 μL of the substrate buffer (1 mL of 1 mg/mL of TMB in DMSO + 9 mL of 0.1 M ammonium citrate buffer + 100 μL of 30% H_2_O_2_) was added. After 30 min, the reaction was stopped via the addition of 100 μL of 2 M sulfuric acid. The absorbance was measured at 450 nm by using a microplate reader. All the assay steps, except for the washing and measurement steps, were performed in the thermoshaker at +37 °C (mixing speed—300 rpm).

White polystyrene plates. Five microliters of Bi-BSA (0.08, 0.4, 2, and 10 μg/mL) in 0.1 M Na-carbonate buffer, pH 9.5, were dotted onto the bottom of white polystyrene plates (Linbro, San Rafael, CA, USA). The plates were kept at +37 °C in a humid chamber for 30 min. The Bi-BSA was removed using a water jet pump. The wells were washed three times with 1 mL of PBS-Tw, and then 500 μL of PB@Gel/Str in PBS-Tw + 1% BSA was added. The plates were incubated at +37 °C in a humid chamber on a shaker for 60 min. After washing, 500 μL of the DAB substrate was added for 5 min. Then, the substrate was removed, and the wells were washed with PBS-Tw.

### 3.7. Direct Assay for Bi-BSA Detection with HRP-Str

One hundred microliters of different concentrations of Bi-BSA (from 100 ng/mL to 1.56 ng/mL) in a 0.2 M carbonate buffer (pH 9.6) were added into the wells of 96-well polystyrene plates. The plates were kept at +4 °C overnight. The plates were washed three times with 350 of PBS with 0.1% Tween-20, pH 7.4 (PBS-Tw) using a microplate washer, and then 200 μL of a blocking buffer (PBS-Tw + 1% casein + 1% BSA, pH 7.4) was added. After 60 min of blocking, the plates were washed three times. The HRP-Str at 1:500 dilution with a blocking buffer (100 μL per well) was added; then, the plates were incubated for 60 min and washed three times with PBS-Tw. After washing, 100 μL of the commercial substrate buffer (Euroimmun, Lübeck, Germany) was added. After 30 min, the reaction was stopped via the addition of 100 μL of 2 M sulfuric acid. The absorbance was measured at 450 nm by using a microplate reader. All the assay steps, except for the washing and measurement steps, were carried out in the thermoshaker at +37 °C (mixing speed—300 rpm).

### 3.8. Western Blotting

Polyacrylamide gels (0.75 mm thickness, 10%) were prepared using 0.1 M Tris–HCl buffer, pH 8.8, with 0.1% SDS. TRIS-glycine buffer (pH ~ 8.3) was used as an electrode buffer. Electrophoresis was performed without using a concentrating gel. Human IgG, mouse monoclonal IgG2a, BSA, or Bi-BSA (all 1 mg/mL) were mixed with a sample buffer (10% SDS, 0.1 M EDTA, 50% glycerol, 0.5 M TRIS, 0.1% Bromphenol blue) in a 5:1 ratio; 10 μL of the mixture was applied to the gel. Electrophoresis was performed at 100 V per gel. Running buffer: TRIS-glycine, pH 8.3–8.5. The gels were cut in half. One half was stained with Coomassie G-250 and then destained with a destaining solution (H_2_O:methanol:acetic acid = 5:4:1). The second half was used for Western blotting. The transferring of proteins from gels onto the nitrocellulose membrane was carried out using the semi-dry method (transfer buffer: 48 mM Tris base, 39 mM glycine, 20% methanol; voltage: 20 mV). Membranes were washed with PBS-Tw (pH 7) and blocked with 1% non-fat dry milk in PBS-Tw overnight +4 °C. After blocking, the membranes were successively treated with 1 μg/mL biotinylated protein A (Bi-PtA) in PBS-Tw (60 min) and 10 μg/mL PB@Gel/Str in PBS-Tw (60 min) at room temperature. The membranes were washed three times with PBS-Tw for 5 min after each incubation step. Then, the membranes were stained using DAB substrate (33 mM Na-citrate with 1.5 M NaCl, pH 6, 1 mg/mL DAB; 0.1% H_2_O_2_). After 5 min, the membranes were washed with a PBS-Tw and dried.

### 3.9. Dot Blot Assay of Antibodies against Spike Protein of SARS-CoV-2

Spike protein, mouse monoclonal IgG2a, Bi-BSA, and BSA (all 2 μL, 0.1 mg/mL in PBS) were spotted onto wet nitrocellulose membrane strips (4 × 50 mm) as described in [40]. After that, the strips were dried for 15 min at room temperature and for 30 min at +37 °C before being washed with 3 mL of PBS-Tw (pH 7) three times for 5 min. Non-specific binding sites were blocked with 3 mL of PBS-Tw + 1% BSA for 60 min. Next, the strips were washed and incubated for 60 min with 3 mL of sera from individuals vaccinated (dilution 1:100, 1:1000, and 1:10,000) and unvaccinated (dilution 1:100) against SARS-CoV-2 diluted in blocking solution. After washing, the test strips were incubated for 60 min in 3 mL of 1 μg/mL Bi-PtA diluted in blocking solution and washed. Then, 10 μg/mL PB@Gel/Str in blocking solution was added for another 60 min. After the final wash, an optimized DAB substrate (33 mM Na-citrate with 1.5 M NaCl, pH 6, 1 mg/mL DAB; 0.1% H_2_O_2_) or commercial DAB substrate (200 mM Na-acetate buffer with 0.54 mg/mL DAB, 1 mM DTPA, 50 mM imidazole, and 0.1% H_2_O_2_) was added and incubated for 5 min. After washing, the test strips were dried and scanned. The brightness and contrast of the obtained images were adjusted to make the colored dots visible on the monitor screen. The brightness and contrast values were the same for all the dot blot images. Two technical replicates were made for each condition.

### 3.10. Immunohistochemistry

Paraffin blocks of the pancreas of healthy Wistar rats were used to assess the efficacy of PB@Gel/Str in tissue staining. The animals were euthanized with 2% xylazine (0.1 mL/100 g body weight) and zoletil (15 mg/100 g body weight). The pancreas was extracted and fixed in 10% neutral formalin. After standard histological processing, pancreatic specimens were embedded in paraffin. Sections (3–5 µm thick) were fixed on X-tra positively charged slides (Leica, Wetzlar, Germany) and used for further immunohistochemical staining. Insulin/Proinsulin primary antibody (Invitrogen, Waltham, MA, USA, MA5-12042) and Biotin Goat Anti-Mouse Ig (Multiple Adsorption) secondary antibody (lot 550337, BD Pharmingen™, Franklin Lakes, NJ, USA) were used. Each incubation step was completed via washing in PBS-Tw, pH 7.4 (0.5% Tween-20). A Leica Bond imaging system (Streptavidin HRP (BD Pharmingen™, 550946) and DAB Substrate Kit (BD Pharmingen™, 550880)) served as a standard treatment. When PB@Gel/Str were used, incubation with the nanoparticles was carried out at room temperature for 30 min in the same way as with Streptavidin HRP. Then, incubation with DAB was carried out for 5 min. Slices were counterstained with hematoxylin and mounted under a coverslip. The optimization experiments used in-house-prepared solutions of DAB (1 mg/mL) in 0.1 M citrate-NaOH buffer (pH 3, 5, and 6) containing 0.1% H_2_O_2_.

The stained histological slides were examined using an Olympus IX-71 optical microscope with a UCMOS03100KPA digital camera, which was used to capture images. Ten photos of islets from each histological slice were used for analysis. Furthermore, the resulting images were segmented (to highlight an islet in the image) using open-source Ilastik software (version 1.4.0.) [41]. The color intensity was measured using open-source CellProfiler software (version 4.2.6.) [42]. A detailed description of the processing pipeline can be found in the Appendix A, and the CellProfiler pipeline file itself, together with example images, can be found via the following link: https://github.com/arteys/PB_image_processing (accessed on 12 May 2023).

## 4. Conclusions

Currently, researchers are conducting extensive studies to optimize the performance of nanozymes in colorimetric assays. The ultimate goal is for them to surpass enzymes in terms of stability and catalytic activity, which would allow for lower detection limits and make test kits less sensitive to storage conditions. To achieve this, various parameters of nanozymes are being adjusted, such as their shape, size, crystal structure, coating chemistry, and doping degree [43]. However, the conditions under which nanozymes operate during the detection step of the assay are rarely taken into account. This study demonstrates that the signal in nanozyme-based colorimetric assays can be increased by simply and cost-effectively optimizing the substrate composition. Specifically, it was demonstrated that buffers like citrate, MES, HEPES, and TRIS containing 1.5–2 M NaCl or NH_4_Cl significantly enhance DAB oxidation by Prussian Blue and provide a higher signal compared to commercial DAB formulations. The most appealing aspect of this approach is its accessibility and versatility. Even a basic screening of commonly available buffers and additives resulted in significant signal enhancement. More advanced approaches, such as the use of ionic liquids [44] or unconventional chemicals, may provide promising ways to further improve the method. An even more exciting option is the optimization of reaction conditions through the application of machine learning [45].

## Figures and Tables

**Figure 1 molecules-28-07622-f001:**
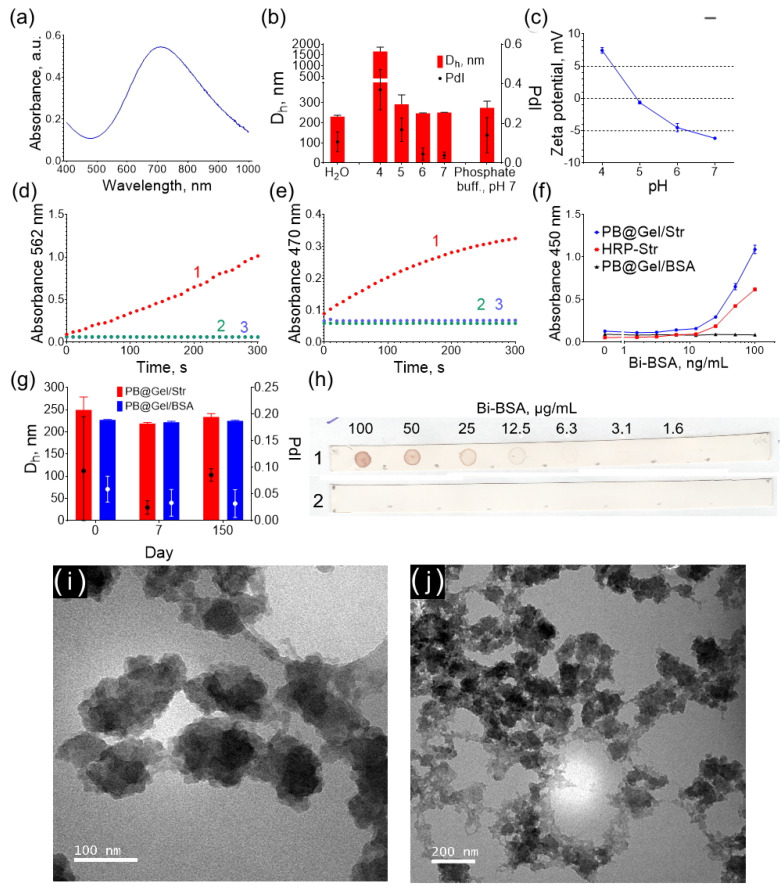
Properties of PB@Gel/Str nanoparticles: (**a**) Absorbance spectrum. (**b**) Size of PB@Gel/Str in water, phosphate buffer, and Na-citrate buffers (pH 4–7). (**c**) Zeta potential at pH 4–7. (**d**) Catalytic activity with TMB substrate (1-PB@Gel/Str + TMB + H_2_O_2_; 2-PB@Gel/Str + TMB; 3-TMB + H_2_O_2_). (**e**) Catalytic activity with DAB substrate (1-PB@Gel/Str + DAB + H_2_O_2_; 2-PB@Gel/Str + DAB; 3-DAB + H_2_O_2_). (**f**) Binding of Bi-BSA (measured via an ELISA-like assay). (**g**) Storage stability of PB@Gel/Str and PB@Gel/BSA. (**h**) Binding of Bi-BSA (measured via a dot blot assay) (1-PB@Gel/Str; 2-PB@Gel/BSA). (**i**,**j**) TEM images of PB@Gel/Str. Mean ± SD, n = 3.

**Figure 2 molecules-28-07622-f002:**
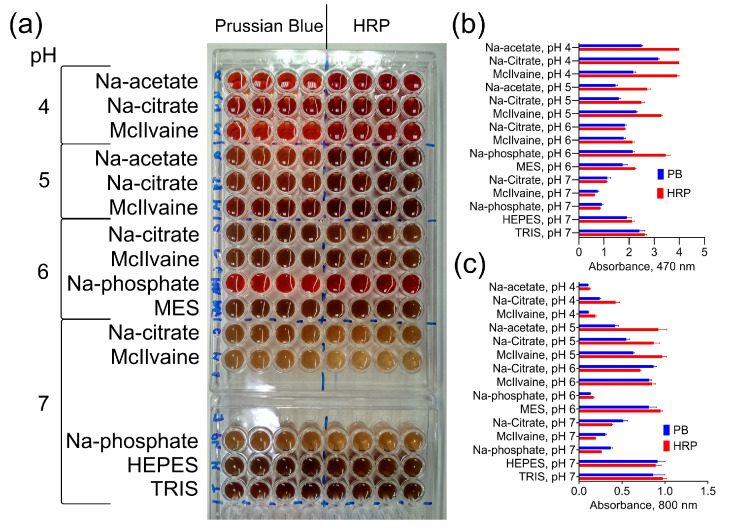
DAB oxidation by PB@Gel/Str and HRP. (**a**) Color of products generated by PB@Gel/Str and HRP. (**b**,**c**) Absorbance of reaction products at 470 and 800 nm. The molarity of all buffers was 33 mM. Nine parts of MES, HEPES, and TRIS–HCl buffers were mixed with one part of Na-citrate buffer with the same pH prior to the experiment. The concentration of H_2_O_2_ was 0.1% for both PB@Gel/Str and HRP. Mean ± SD, n = 4.

**Figure 3 molecules-28-07622-f003:**
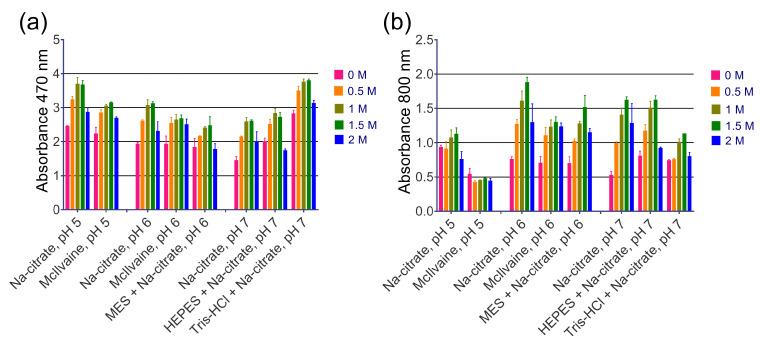
The effect of NaCl concentration on the intensity of DAB oxidation. (**a**) Absorbance of reaction products at 470 nm. (**b**) Absorbance of reaction products at 800 nm. The final concentrations of NaCl in the substrate are given. The molarity of all buffers was 33 mM. Nine parts of MES, HEPES, and Tris–HCl buffers were mixed with one part of Na-citrate buffer with the same pH prior to the experiment. Mean ± SD, n = 2.

**Figure 4 molecules-28-07622-f004:**
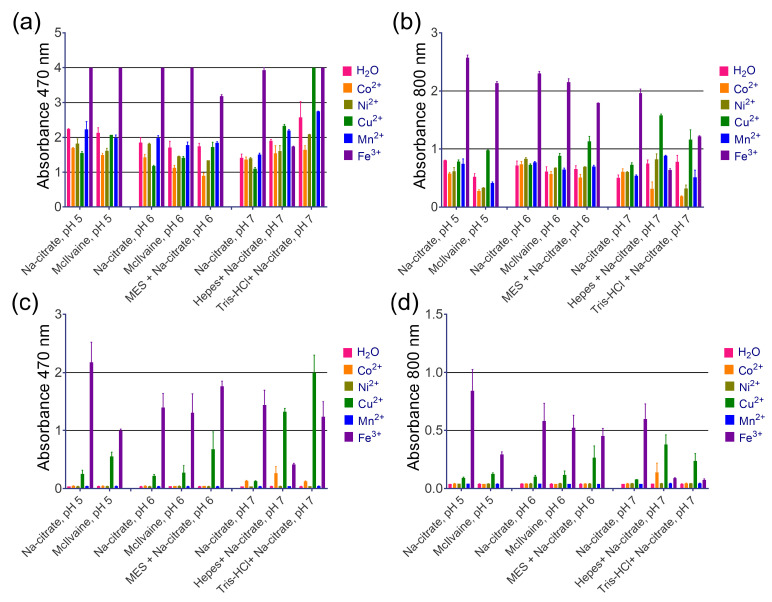
The effect of metal cations on the intensity of DAB oxidation. (**a**,**b**) Absorbance of substrate at 470 and 800 nm after the addition of PB@Gel/Str. (**c**,**d**) Absorbance of substrate at 470 and 800 nm before the addition of PB@Gel/Str. Mean ± SD, n = 2. The final concentration of Me^2+/3+^ in the substrate was 0.8 mM. The molarity of all buffers was 33 mM. Nine parts of MES, HEPES, and Tris–HCl buffers were mixed with one part of Na-citrate buffer with the same pH prior to the experiment.

**Figure 5 molecules-28-07622-f005:**
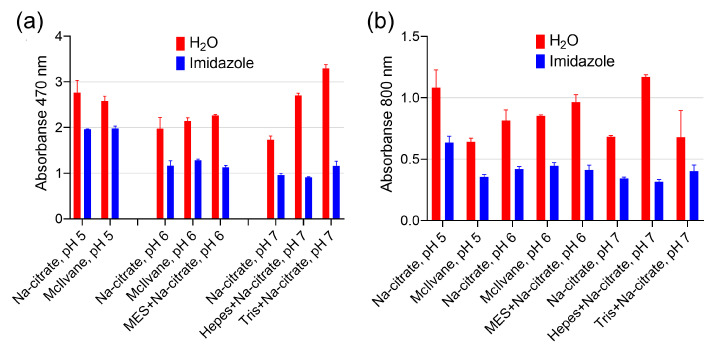
The effect of imidazole on the intensity of DAB oxidation. (**a**) Absorbance of reaction products at 470 nm. (**b**) Absorbance of reaction products at 800 nm. The final concentration of imidazole was 10 mM. Before being added to the buffer, the pH of the imidazole solution was adjusted using HCl to match the pH of the buffer. The molarity of all buffers was 33 mM. Nine parts of MES, HEPES, and Tris–HCl buffers were mixed with one part of Na-citrate buffer with the same pH prior to the experiment. Mean ± SD, n = 2.

**Figure 6 molecules-28-07622-f006:**
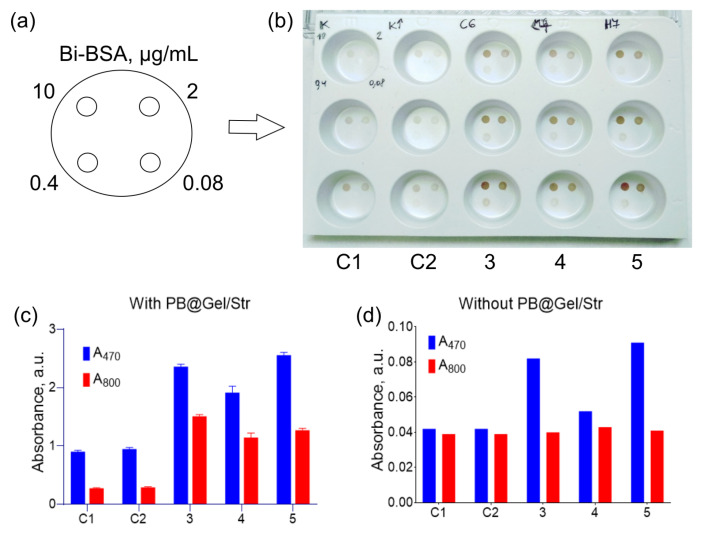
Comparison of optimized DAB substrates with commercial formulations via dot blot assay (**a**,**b**) and in 96-well plates (**c**,**d**). (**a**) Location of Bi-BSA spots in the wells. (**b**) Direct Bi-BSA detection by PB@Gel/Str in white polystyrene plates. (**c**) Absorbance of substrates at 470 and 800 nm after the addition of PB@Gel/Str. (**d**) Absorbance of substrates at 470 and 800 nm in the absence of nanozymes. Substrates for (**b**–**d**): C1–200 mM Na-acetate buffer with 0.54 mg/mL DAB, 1 mM DTPA, 50 mM imidazole, and 0.03% H_2_O_2_, pH 5; C2–200 mM Na-acetate buffer with 0.54 mg/mL DAB, 1 mM DTPA, 50 mM imidazole, and 0.1% H_2_O_2_, pH 5; 3–33 mM Na-citrate buffer with 1.5 M NaCl, pH 6, 1 mg/mL DAB, 0.1% H_2_O_2_; 4–33 mM Na-citrate buffer with 1.5 M NH_4_Cl, pH 7, 1 mg/mL DAB, 0.1% H_2_O_2_; 5–30 mM HEPES–HCl buffer with 3 mM Na-citrate and 1.5 M NaCl, pH 7, 1 mg/mL DAB, 0.1% H_2_O_2_. Mean ± SD, n = 3.

**Figure 7 molecules-28-07622-f007:**
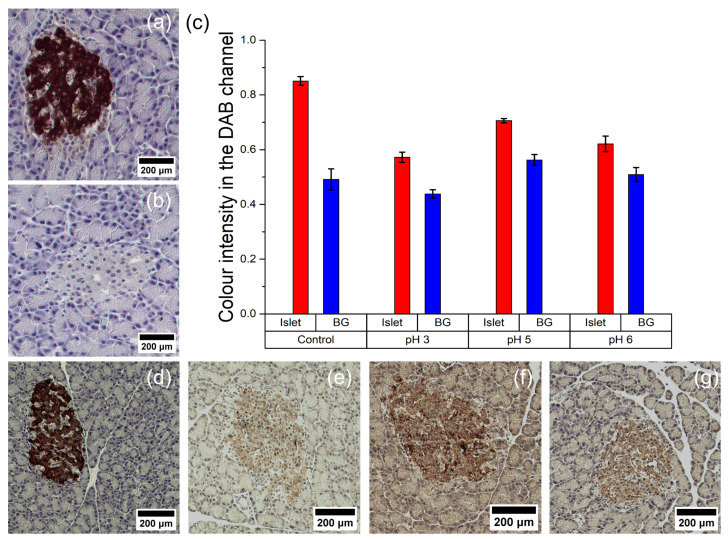
Immunohistochemical staining of pancreatic islets with horseradish peroxidase (**a**,**d**) and Prussian Blue nanoparticles (**b**,**e**–**g**). (**c**) Data on staining intensity obtained after digital processing of histological images taken with different DAB solutions (BG—background). Commercial DAB solution (**a**–**d**) or 100 mM Na-citrate buffer with 1 mg/mL DAB, 1 and 0.1% H_2_O_2_, pH 3 (**e**), pH 5 (**f**), or pH 6 (**g**) were used for staining.

**Figure 8 molecules-28-07622-f008:**
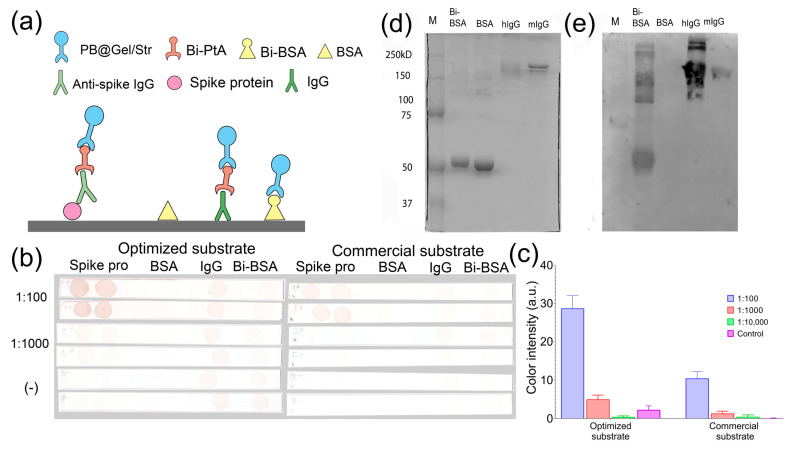
(**a**–**c**) Dot blot assay of IgG against SARS-CoV-2 spike protein (two spots of spike protein) in pooled human sera obtained from seronegative (−) individuals (dilution 1:100) and seropositive (+) individuals (dilution 1:100 and 1:1000). Bi-BSA and hIgG were used as positive controls, while BSA served as the negative control (one spot for each). (**a**) Schemes of the assay are provided for each spot. (**b**) Test strips. (**c**) Quantitative analysis of spot intensity. (**d**) SDS-PAGE of BSA, Bi-BSA, IgG from human serum (hIgG), and mouse monoclonal IgG2a (mIgG). Coomassie Brilliant blue staining. The molecular weight values of protein markers are given. (**e**) Western blotting of the same samples.

**Table 1 molecules-28-07622-t001:** Preparation of substrate solutions in optimization experiments.

Reagent	Volume, μL
Water	150
Buffer	100
2% H_2_O_2_	15
10 mg/mL DAB in H_2_O	30
Nanozymes, 120 μg/mL	5
Total volume	300

## Data Availability

Data are contained within the article and Appendix A.

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
