# Peer review of "Optimizing the Composition of the Substrate Enhances the Performance of Peroxidase-like Nanozymes in Colorimetric Assays: A Case Study of Prussian Blue and 3,3′-Diaminobenzidine"

_molecules, 2023, doi:10.3390/molecules28227622_

Round 1

Reviewer 1 Report

Comments and Suggestions for Authors

Comments:
In this work, the authors present a step-by-step optimization of the DAB substrate composition for the Prussian Blue nanozymes. They demonstrate that the optimized substrate outperforms commercial formulation in terms of efficiency. However, the results are all phenomenon report without any discussion in depth. The scientific impact should be low. Importantly, the English writing and quality of this manuscript need to be improved. Therefore, I do not recommend this paper for publication at this stage.

Comments on the Quality of English Language

Extensive editing of English language required.

Author Response

The authors of the article would like to express their gratitude to the respected reviewer for their valuable comments. Below, you can find the responses to them. All the changes in the article's text are highlighted in yellow. We hope that the revised manuscript is suitable for publication.

  1. In this work, the authors present a step-by-step optimization of the DAB substrate composition for the Prussian Blue nanozymes. They demonstrate that the optimized substrate outperforms commercial formulation in terms of efficiency. However, the results are all phenomenon report without any discussion in depth. The scientific impact should be low. Importantly, the English writing and quality of this manuscript need to be improved. Therefore, I do not recommend this paper for publication at this stage.

Response

The authors fully agree with the reviewer that this paper lacks mechanistic details regarding the effects of substrate composition.

Initially, our primary objective was to demonstrate the application of Prussian Blue nanozymes as peroxidase mimics in immunochemical techniques such as western blot and immunohistochemistry, where these nanozymes have not been utilized previously. However, during the course of our study, we noticed that the signal intensity of the nanozyme was very low when we used substrate compositions optimized for peroxidase. As a result, we explored different substrate compositions and discovered that the DAB substrates optimized for peroxidase may not be optimal for the nanozyme.

In an attempt to gain further insight, we contacted the authors of papers focusing on the application of nanozymes in tissue staining and inquired about their substrate choices. They responded by stating that the substrate composition is an insignificant technical detail and that they used a commercial DAB substrate optimized for peroxidase. Consequently, we decided to prioritize the optimization of the DAB substrate in order to demonstrate its potential benefits in terms of assay performance. We believe that this message holds significant value, even as a standalone paper. Understanding the mechanisms involved is a goal for our future research.

  1. Extensive editing of English language required.

Response

The manuscript has been revised and spell-checked based on the referee's comments.

Reviewer 2 Report

Comments and Suggestions for Authors

I have carefully examined manuscript Optimizing the composition of the substrate enhances the per-2 formance of peroxidase-like nanozymes in colorimetric assays: 3 a case study of Prussian Blue and 3,3’-diaminobenzidine submitted to Molecules journal. Tha manuscript contains interesting data, is nice planed and written. Further, presented data might be use in practical applucation. I have several concerns about this manuscript. See my details comments below.

My main concerns is related to the fact that Authors present study using animal and human tissues. For this kind of study necessary consents and authorizations are required. Do the Authors have required permissions? If so, please present them in the manuscript.

Some of the most important results should be presented in Abstract.

There is no deeper discussion on results presented in Fig. 4. Please improve.

There is laso lack of deeper discussion on results presented in Fig. 8.

Section 3.5. Provide the detailed ranges for each parameter tested in ptimization study.

Improve all editorial, grammar and language issues presented in the manuscript.

Crucial results/findings obtained in the study should be presented in the Conclusions to support presented remarks.

Comments on the Quality of English Language

Crucial results/findings obtained in the study should be presented in the Conclusions to support presented remarks.

Author Response

The authors of the article would like to express their gratitude to the respected reviewer for their valuable comments. Below, you can find the responses to them. All the changes in the article's text are highlighted in yellow. We hope that the revised manuscript is suitable for publication.

  1. My main concerns is related to the fact that Authors present study using animal and human tissues. For this kind of study necessary consents and authorizations are required. Do the Authors have required permissions? If so, please present them in the manuscript.

Response

"The information regarding the authorization for experiments involving animal and human tissues can be found in the relevant section of the manuscript, situated above the References section (please see below). Both human sera and rat pancreases were obtained during our previous research. Human sera were sourced from the collection of the Institute of Ecology and Genetics of Microorganisms. Paraffin blocks containing normal rat pancreas were generously provided by the laboratory of morphology and biochemistry at the Institute of Immunology and Physiology of RAS. Approval from the Local Ethical Committee was obtained prior to commencing the experiments.

“Institutional Review Board Statement: This research was performed according to World Medical Association’s Declaration of Helsinki and Council of Europe Protocol to the Convention on Human Rights and Biomedicine and approved by the Ethics Committee of the Institute of Ecol-ogy and Genetics of Microorganisms, Ural Branch of the Russian Academy of Sciences (IRB00010009).”

  1. Some of the most important results should be presented in Abstract.

Response

The following sentences have been added: “Through the use of 3,3'-diaminobenzidine (DAB) and Prussian Blue as a model chromogenic substrate and nanozyme, we illustrate that the presence of enhancers, such as imidazole, in commercial substrates diminishes the catalytic activity of nanozymes. Conversely, a simple modification of the substrate buffer greatly enhances the performance of nanozymes. Specifically, we demonstrate that buffers such as citrate, MES, HEPES, and TRIS, containing 1.5-2 M NaCl or NH4Cl, substantially increase DAB oxidation by Prussian Blue and yield a higher signal compared to commercial DAB formulations.”

  1. There is no deeper discussion on results presented in Fig. 4. Please improve.

Response

The following discussion has been included: “We selected several cations (Fe3+, Mn2+, Co2+, Cu2+, Ni2+) based on the literature. Cupric and ferric cations were observed to increase absorbance at 470 and 800 nm, as shown in Figure 4a and 4b. However, this effect can be attributed to their inherent peroxidase-like activity, as the formation of a brown product was observed even before the addition of nanozymes (spontaneous DAB oxidation), as depicted in Figure 4c and 4d.”

  1. There is laso lack of deeper discussion on results presented in Fig. 8.

Response

The following discussion has been included: “In the positive sample, brown spots were observed in the areas of the membrane where the spike protein was adsorbed, as shown in Figure 8b. In contrast, no spots were observed on the test strip incubated in the negative sample, as depicted in Figure 8c. Quantitative analysis revealed an almost three-fold increase in spot coloration in samples treated with the optimized substrate compared to the commercial substrate used in Section 3.2. This finding confirms the efficacy of the optimization procedure.”

  1. Section 3.5. Provide the detailed ranges for each parameter tested in ptimization study.

Response

Ranges for all tested parameters have been added to Table S2.

  1. Improve all editorial, grammar and language issues presented in the manuscript.

Response

The manuscript has been revised and spell-checked based on the referee's comments.

  1. Crucial results/findings obtained in the study should be presented in the Conclusions to support presented remarks.

Response

The following sentence has been added: “Specifically, we showed that buffers like citrate, MES, HEPES, and TRIS containing 1.5-2 M NaCl or NH4Cl significantly enhance DAB oxidation by Prussian Blue and provide a higher signal compared to commercial DAB formulations.”

Reviewer 3 Report

Comments and Suggestions for Authors

This work presented a step-by-step idea for optimizing the composition of DAB substrates for Prussian Blue nanozymes, demonstrating that the optimized substrates outperform commercial preparations in terms of efficiency. Also, this work applied the optimized DAB substrate in nanozyme based immunostaining techniques, especially in Western blotting assays of mouse and human IgG and in spot blotting assays of human antibodies against the SARS-CoV-2 spiking protein. This is the first application of Prussian Blue nanozymes as peroxidase-like markers in biochemistry and protein blotting. Overall, this work is interesting. Some minor modifications are needed to further improve this work.

1. Optimizing the composition of the substrate can be an efficient and cost-effective way to improve the defined detection limit. However, this work seems to lack some overview of the relevant advances.

2. Measurement of the PB@Gel/Str sample by DLS yielded a relatively narrow size distribution, which was seen to be more homogeneous. However, TEM analysis shows that the shape is extremely irregular. This may require explanation.

3. In MES and sodium citrate buffers at pH 6 and 7, an increase in DAB concentration to 2 or 3 mg/mL resulted in higher absorbance, and why was there no significant effect on the background signal?

4. Why is the zeta potential of PB@Gel/Str negative below pH 5? This needs to be explained.

5. In Figure S4, the adsorption of biotinylated and non-biotinylated oligo DNA by PB@Gel/BSA is extremely different. This may require explanation.

6. In Figure 7B, in the case of nanoenzymes, yet no signal is detected. What is the reason for this?

7. In Figure 7C, why does the intensity of staining decrease at pH greater than 5?

Comments on the Quality of English Language

Minor editing of English language required

Author Response

The authors of the article would like to express their gratitude to the respected reviewer for their valuable comments. Below, you can find the responses to them. All the changes in the article's text are highlighted in yellow. We hope that the revised manuscript is suitable for publication.

This work presented a step-by-step idea for optimizing the composition of DAB substrates for Prussian Blue nanozymes, demonstrating that the optimized substrates outperform commercial preparations in terms of efficiency. Also, this work applied the optimized DAB substrate in nanozyme based immunostaining techniques, especially in Western blotting assays of mouse and human IgG and in spot blotting assays of human antibodies against the SARS-CoV-2 spiking protein. This is the first application of Prussian Blue nanozymes as peroxidase-like markers in biochemistry and protein blotting. Overall, this work is interesting. Some minor modifications are needed to further improve this work.

  1. Optimizing the composition of the substrate can be an efficient and cost-effective way to improve the defined detection limit. However, this work seems to lack some overview of the relevant advances.

Response

Despite the existence of numerous reports on the impact of small molecules and ions on nanozyme activity, the use of acetate buffer for the preparation of nanozyme substrates is predominant in current literature. In this study, we have presented several examples of substrate composition optimization for various colorimetric assays. However, it should be noted that these optimizations are still limited to a relatively small number of buffers, typically ranging from 3 to 5.

The following text was added into the Introduction section: “Some of these reports are summarized below. Hormozi-Jangi et al. demonstrated significantly higher efficiency of the DAB substrate prepared using an acetate buffer compared to citrate, TRIS, and phosphate buffers in the MnO2-based assay of triacetone triperoxide. In contrast, another study indicated that the effect of the acetate buffer was nearly identical to that of the phosphate and citrate buffers in the V2O5-based colorimetric assay. The influence of the buffer was also observed in lac-case-mimicking copper-based nanozymes utilized for phenol identification”

  1. Measurement of the PB@Gel/Str sample by DLS yielded a relatively narrow size distribution, which was seen to be more homogeneous. However, TEM analysis shows that the shape is extremely irregular. This may require explanation.

Response

The following explanation was added: “This is likely due to the well-known bias of dynamic light scattering (DLS) towards detecting larger particles. Smaller particles scatter light less strongly, making them more challenging to detect and not significantly impacting the polydispersity index value. Additionally, the DLS method assumes that the measured objects are spherical in shape. For non-spherical objects with specific diffusion coefficients, the method provides the size of an equivalent sphere. As a result, the obtained hydrodynamic diameters are larger than the actual size of the nanoparticles but still offer a relatively accurate estimate of their size.”

  1. In MES and sodium citrate buffers at pH 6 and 7, an increase in DAB concentration to 2 or 3 mg/mL resulted in higher absorbance, and why was there no significant effect on the background signal?

Response:

During the optimization experiments, the term "background signal" referred to the absorbance of the substrate before the addition of nanozymes. This absorbance was attributed to the spontaneous oxidation of DAB by hydrogen peroxide. In certain conditions, this oxidation was significant, potentially due to the presence of trace amounts of transition metals in buffers and salts. This phenomenon was particularly evident in the experiments involving the addition of cations. Higher absorbance values observed after the addition of nanozymes indicate the effectiveness of the substrate, which generates a strong signal upon reaction with the nanozymes, while minimizing the spontaneous oxidation of DAB.

In order to avoid confusion with non-specific staining of test-strips and tissue slices, we have replaced the terms "background" and "background signal" with "spontaneous oxidation" when referring to the optimization of DAB substrates.

The mechanistic aspect of this phenomenon is currently unknown. It is possible that these two buffers, citrate and MES, have the ability to suppress the oxidative effects of metal traces. This is not surprising in the case of citrate buffer, as it is known to possess chelating properties towards cations. However, MES buffer has low chelating ability, making it unlikely to operate through the same mechanism.

  1. Why is the zeta potential of PB@Gel/Str negative below pH 5? This needs to be explained.

Response:

The following explanation has been added to the section: 2.1: “The zeta potential of PB@Gel/Str was negative above pH 5 (Figure 1(c)). The zeta potential of the nanoparticles is influenced by the isoelectric point of gelatin A, which serves as the coating. The isoelectric point of gelatin A typically falls within the pH range of 7 to 9. However, treatment with glutaraldehyde removes lysine amino groups from the gelatin layer, shifting the isoelectric point of the aldehyde-treated gelatin A to a pH range of 4 to 5.”

  1. In Figure S4, the adsorption of biotinylated and non-biotinylated oligo DNA by PB@Gel/BSA is extremely different. This may require explanation.

Response:

In our experiments on oligoDNA adsorption, we observed significantly higher adsorption of the amino-terminated aptamer compared to the biotinylated aptamer on PB@Gel/BSA. This suggests that the presence of a terminal amino group somehow influences DNA adsorption. One potential mechanism is the interaction of the terminal amine with unquenched aldehyde groups that may be present in the nanoparticles following synthesis. However, this scenario seems unlikely since the binding buffer contained a significant amount of BSA and casein (1%), which should have quenched any remaining carbonyl groups. It is possible that the small size of the oligoDNA allows it to react with the aldehyde groups that are not accessible to proteins due to steric hindrance. We do not believe that electrostatic interactions play a significant role in this process, as both the oligoDNA (even the amino-terminated one) and the nanoparticles are negatively charged at pH 7.

  1. In Figure 7B, in the case of nanoenzymes, yet no signal is detected. What is the reason for this?

Response:

In this experiment, we conducted staining of pancreatic islets using polyHRP and Prussian Blue nanozymes. For both cases, we utilized a commercial DAB substrate solution. However, this substrate solution proved to be ineffective in the case of nanozymes due to its suboptimal composition and reduced concentration of hydrogen peroxide. Corresponding clarification is given in the Section 2.3 (highlighted in yellow): “It provided intense staining for HRP (Figure 7(a)), but no signal was detected in the case of the nanozyme (Figure 7(b)). Possible reasons for this could be the presence of quenching additives or suboptimal hydrogen peroxide concentration. In the second experiment (Fig-ure 7(d)-(g)), sections treated with PB@Gel/Str were incubated in a custom DAB solution based on Na-citrate buffer without imidazole”

  1. In Figure 7C, why does the intensity of staining decrease at pH greater than 5?

Response:

In this experiment, we stained pancreatic islets using Prussian Blue and a DAB substrate based on sodium citrate buffers with pH values of 3, 5, and 6. With this type of buffer, at lower pH values (3-4), the oxidation of DAB is highly efficient, but insoluble polymers of oxidized DAB are not formed, resulting in a pale brown staining. At more neutral pH values (6-7), the oxidation efficiency decreases, but the oxidized DAB readily forms polymers. At pH 5, both processes (oxidation of DAB and formation of polymeric species) are intense. Consequently, staining of the islets at pH values higher or lower than 5 exhibits reduced color intensity.

Round 2

Reviewer 1 Report

Comments and Suggestions for Authors

In brief, the experimental design was not reasonable and then less new information and data were showed in the main text. For example, Figure 2 shows well-known common phenomena because peroxidase substrates are typically pH dependent (usually at pH < 5). At line 186, authors mentioned that “However, peroxidase generated higher absorbance values at pH 4 and 5, but not at pH 6 and 7.” Is this new finding? In addition, there was no comment or conclusive results of Figure 3. Moreover, at line 240, “Cupric and ferric cations were observed to increase absorbance at 470 and 800 nm, as shown in Figure 4a and 4b. However, this effect can be attributed to their inherent peroxidase-like activity,…”. This indicated that the data in Figure 4 was all predictable without any variable information. Therefore, I do not recommend this paper for publication.

Comments on the Quality of English Language

After revision, there are still some incomprehensible sentences.

At Line 153, “The binding of non-biotinylated oligoDNA was 8-fold lower (Figure S4)”.

At Line 175, “Prussian Blue nanozymes are first reduced to Prussian White by TMB, and only after that are they oxidized by H2O2.”

At Line 209, “Three of the most promising substrates were chosen for comparison with commercial formulations.”

At Line 301, “The optimized substrates showed significantly enhanced color intensity in the dot blot assay of Bi-BSA compared to the control substrates (Figure 6(a),(b)), and there was no non-specific staining of the polystyrene surface, despite a slight degree of spontaneous DAB oxidation (Figure 6(d)).”

At Line 304, “Comparison of the substrates in 96-well plates confirmed the results of the dot blot assay, with the optimized substrates generating 2-5-fold higher absorbance values at both 470 and 800 nm (Figure 6(c)).”

At Line 324, “This resulted in reliable staining of the islets, which were clearly distinguishable from the surrounding tissue both to the naked eye (Figure 7(e)-(g)) and through image analysis software (Figure 7(c)).”

At Line 341, “In previous sections, we performed a simplified version of dot blot for the characterization of Prussian Blue conjugates and substrate comparison.”

At Line 352, “Protein A binds to both human and mouse antibodies[38], and the biotin tag facilitates its interaction with PB@Gel/Str.”

Author Response

Dear Reviewer,

We would like to express our sincere gratitude for your valuable feedback and time spent reviewing our manuscript. Your insightful comments have been truly appreciated. We have made the necessary revisions to the English language in accordance with the reviewer's recommendations. The modified sections of the text are highlighted in blue. We hope that the quality of the article now meets the standards of the journal.

Thank you once again for your constructive input.

Sincerely,
Pavel Khramtsov

In brief, the experimental design was not reasonable and then less new information and data were showed in the main text. For example, Figure 2 shows well-known common phenomena because peroxidase substrates are typically pH dependent (usually at pH < 5). At line 186, authors mentioned that “However, peroxidase generated higher absorbance values at pH 4 and 5, but not at pH 6 and 7.” Is this new finding?

Response

The pH optimum for peroxidase varies depending on the substrate. For instance, with TMB, the maximum signal is typically achieved at pH 4-6, while with DAB, the relationship between pH and substrate conversion is more complex. At lower pH levels (4-5), the oxidation of DAB is highly efficient, but it does not form the polymeric species necessary for creating colored spots in blotting or immunohistochemistry. At more neutral pH levels, the polymerization of DAB is more prominent, resulting in the formation of brown spots.

Figure 2 illustrates the comparison between HRP and Prussian Blue. The text states "First, we compared the catalytic activity of Prussian Blue and HRP at various pH levels in different buffers. The oxidation degree was measured by monitoring the absorbance at 470 nm, while absorbance at 800 nm (turbidity) indicated the formation of DAB polymer species. In general, the effect of buffer composition on DAB oxidation was similar for both catalysts (Figure 2). However, peroxidase generated higher absorbance values at pH 4 and 5, but not at pH 6 and 7."

Our intention was to convey that peroxidase exhibits higher absorbance values than Prussian Blue at pH 4 and 5, but not at pH 6 and 7. This observation highlights the significance of selecting the appropriate buffer for nanozyme-based immunoassays. Unfortunately, this aspect is often overlooked in the development of such assays. Many authors of nanozyme-related papers utilize buffers that are optimized for peroxidase, thereby missing the opportunity to achieve a better signal.

To enhance clarity, the revised version of this sentence will read as follows: "However, peroxidase generated higher absorbance values than Prussian Blue at pH 4 and 5, but not at pH 6 and 7."

2. In addition, there was no comment or conclusive results of Figure 3.

Response

The discussion of the results presented in Figure 3 can be found in various sections of the manuscript.

  1. Line 217:  "The addition of NaCl or NH4Cl up to 1.5 M resulted in enhanced A470 and A800 for most of the tested buffers, while KCl had no effect (Figure 3, Figure S5, S6, S7)"
  2.  Lines 255-261 : "Overall, the addition of salts proved to be the most effective method for enhancing the signal. The enhanced effect of high salt concentration could be attributed to a salting-out effect or a lowering of the buffer pH [37]. We tested the effect of salt at various pH levels of the same buffer and found that the change in pH did not result in increased turbidity, as observed with salt addition. Therefore, we believe that the salting-out effect, rather than a shift in pH caused by high salt concentration, is responsible for the stronger signal. Salts can also affect the substrate's affinity towards the catalyst or accelerate DAB polymerization/aggregation, leading to higher turbidity values (A800)"

3. Moreover, at line 240, “Cupric and ferric cations were observed to increase absorbance at 470 and 800 nm, as shown in Figure 4a and 4b. However, this effect can be attributed to their inherent peroxidase-like activity,…”. This indicated that the data in Figure 4 was all predictable without any variable information. Therefore, I do not recommend this paper for publication.

Response

We disagree with the notion that the results concerning the impact of metal cations were predictable. The tested transition metal cations, which are well-known signal enhancers in HRP-based assays (primarily Ni2+), had no significant effect on the activity of Prussian Blue nanozymes. It should be noted that we used cation concentrations from the literature and did not explore concentrations other than 0.8 mM.

The oxidation of DAB initiated by Cu2+ and Fe3+ cations themselves (without the addition of Prussian Blue) was possible, as many transition metal cations possess some peroxidase-like activity, but this outcome was not foreseeable. We introduced metal cations into various substrate buffers to assess whether they would enhance the signal in the Prussian Blue-based assay. Only two of them produced a higher signal: Cu2+ and Fe3+.

An important factor to consider is that nanozymes require a higher concentration of hydrogen peroxide. In our study, the concentration of H2O2 in the substrate buffer was higher than what is typically required for peroxidase. At such elevated levels of H2O2, traces of metal present in the buffers can initiate the oxidation of DAB. All the substrate buffers contained citrate anions that chelate metal cations and suppress their ability to oxidize DAB. However, in the case of Cu2+ and Fe3+, the chelating effect of citrate was not strong enough, allowing these cations to oxidize DAB regardless. Control experiments (measuring the absorbance of the substrate buffer without the addition of Prussian Blue) were conducted and confirmed that the signal enhancement was not due to the effect of cations on nanozyme activity, but rather the oxidation of DAB by the cations themselves.

Reviewer 2 Report

Comments and Suggestions for Authors

I have examined the revised manuscript and Authors response to my questions. In my opinion, the quality of the manuscript was improved upon revision and deeper explanantions of the obtaine data were added. Authors have also answered to my questions an dprovided permissions required to work on human and animal cells. In my opinion, manuscript might be accepted in the present form.

Comments on the Quality of English Language

English language used in the maniscript is generally fine. Onlu some minor improvements could be done.

Author Response

Dear Reviewer,

We would like to express our sincere gratitude for your valuable feedback and time spent reviewing our manuscript. We have made the necessary revisions to the English language in accordance with the reviewer's recommendations. 

Sincerely,
Pavel Khramtsov